# FEDBIOT: A SOLUTION FOR FEDERATED LARGE LANGUAGE MODEL FINE-TUNING WITH INTELLECTUAL PROPERTY PROTECTION

## ABSTRACT

Due to data and information privacy concerns, data owners are not willing to share the data with others, but each of them may not have sufficient data to fine-tune a satisfactory large language model (LLM) individually. Parallelly, the LLM owners may not be willing to disclose the LLMs' details, including their architectures and parameters. Therefore, this leads to the challenge of fine-tuning an LLM on a federated learning task where the clients with task-specific data cannot obtain the complete LLM. To solve the challenge, this paper introduces FedBiOT, a method that guarantees the clients' data privacy and avoids the disclosure of an LLM. Specifically, we formulate and solve a bi-level optimization problem to ensure that the emulator distilled on the public dataset by the LLM owner can help the adaptors' local fine-tuning on clients' private datasets, regardless of the distribution drift between those datasets. Different clients' adapters are synchronized in a federated learning style, and the full model composed with the final derived adapter can achieve better performance on downstream tasks. We conduct extensive experiments on LLaMA-7B training for various federated learning tasks and witness significant improvements over existing baselines.

## 1 INTRODUCTION

The recent large language models (LLMs) (OpenAI, 2023b; Touvron et al., 2023; Scao et al., 2022; Zeng et al., 2023; Chowdhery et al., 2022) have achieved incredible performance of text generation, solving a wide range of natural language tasks, such as code generation (Chen et al., 2021) and math problem-solving (Wei et al., 2022). When it comes to a specific task, the model could perform better after being enriched by the knowledge introduced by fine-tuning (Hu et al., 2021; Lester et al., 2021). The quality and quantity of the task-specific data are directly related to the performance of the fine-tuned model on downstream tasks: large and well-labeled data can significantly improve the model, while small and irrelevant data can only benefit the model marginally. However, there are many cases where task-specific data are possessed by multiple data parties, while each of them may have a limited number of samples that can be used to fine-tune LLMs. For example, a hospital in a rural area may only have a limited number of lung cancer cases recorded in its own system; if an LLM is only fine-tuned on one set of those cases, it may not obtain comprehensive knowledge and easily be overfitted.

Therefore, an effective way is to collect the data from multiple clients, but this is confronted with threefold challenges:

- **Clients' limited computation capability.** Besides the limited data challenges, the downstream clients usually own limited computation power, so they prefer an eco-efficient method to achieve their goals. However, fine-tuning LLMs can be more computationally expensive than traditional machine-learning tasks because traditional model sizes cannot be comparable with LLMs'.

- **Privacy of clients' data.** Although there are LLM owners offering fine-tuning APIs as services, the users must pack their data as files and upload them to use a black-box fine-tuning (OpenAI, 2023a). However, some privacy-sensitive businesses may not accept those API services. Some internal data are directly related to trade secrets, such as the codebase of a technology company or the paperwork of a law firm. More generally, data from customers have restricted usage by data

privacy regulations (GDPR, 2016; CCPA, 2023), so their local data cannot be directly shared with any other parties without customers' consent.

• **Intellectual property of LLMs.** Pre-training an LLM is very costly, so the LLMs are generally protected under different protocols. From the perspective of intellectual property protection, the powerful LLMs can be categorized as open-source or closed-source. For the open-source LLMs, although the public can download the models with pre-trained parameters to build downstream applications or conduct research based on them, the applications based on those models are still restricted by open-source protocols. For example, the open-source protocol may prevent the application using the open-source LLM from commercialization or require the application to be open-source hereditarily. On the contrary, the public cannot access the closed-source model architecture or parameters, while only the inference or fine-tuning APIs are provided. Moreover, the best closed-source LLMs still have leading performance on a wide range of language tasks, making the public believe its leading edge can be maintained or even enhanced after fine-tuning.

This paper focuses on reconciling the *data protection compulsoriness* of the downstream fine-tuned model users and the *need for intellectual property protection* from the closed-source LLM owners, with a *computationally efficient algorithm*. A closely related work is offsite-tuning (Xiao et al., 2023), which focuses on cases with only one fine-tuning client. We share similar goals on data privacy protection and LLM intellectual property protection, but we extend the problem to the federated learning (FL) setting. In our setting, we assume that the LLM owner does not collect data directly from clients (OpenAI, 2023a) but serves as the server in FL, who can use a public dataset to distill her LLM and aggregate some local updates on part of the model from clients; multiple clients want to collaborate on fine-tuning for similar downstream tasks (e.g., code completion). Moreover, following the classic FL setting, we do not assume the data distribution on clients or the public data owned by the server to be the same. For example, a client's codebase can mainly consist of C++ code, while the other's main programming language is Python. Our goal, in general, is to provide a framework for the collaborative clients to fine-tune without access to full LLM nor sharing local data directly, but the fine-tuned model can still achieve better performance than fine-tuning LLM locally with their local data exclusively.

**Contribution.** We first revisit the offsite-tuning (Xiao et al., 2023) and its federated version (Kuang et al., 2023), and identify the prerequisites of achieving data privacy and model intellectual property protection in the federated learning setting. Then, we identify the limitations when these two algorithms are applied in the FL setting: when the public dataset for model distillation has a very different data distribution from that of the clients' datasets, the distilled emulator cannot faithfully simulate the full model's behavior on the clients' datasets. Consequentially, federated fine-tuning with such an emulator may not lead to satisfactory performance. To overcome this issue, we formulate the problem as a bi-level optimization task, **Fed**erated **Bi**-level **O**ffsite **T**uning (FedBiOT). Our solution to the optimization problem can achieve 1) friendliness for computation-limited clients, 2) reconciliation of the data privacy and model intellectual property protection, and 3) balance on the impact of the distillation dataset and clients' fine-tuning dataset in the training process. We conduct the experiments on LLaMA for fine-tuning with three tasks, i.e., code generating, math problem solving, and question answering. The proposed approach has significant improvement over all these tasks compared with the baseline approaches. We also empirically identify the effectiveness of each sub-component of the algorithm via a set of ablation studies, providing insights for potential improvement in the future.

## 2    MOTIVATIONS

Consider there is an FL system with a total of $M$ clients, denoted by $[M]$. Each client $m \in [M]$ holds a local dataset $\mathcal{D}_m$. A client's local loss is defined as $F_m(\boldsymbol{w}) := \frac{1}{|\mathcal{D}_m|} \sum_{(\boldsymbol{x},\boldsymbol{y}) \in \mathcal{D}_m} f\left(\mathcal{M}(\boldsymbol{x}; \boldsymbol{w}); \boldsymbol{y}\right)$, where $\mathcal{M}(\boldsymbol{x}; \boldsymbol{w})$ is the output on a given model parameterized by $\boldsymbol{w}$ and an input $\boldsymbol{x}$. The loss function $f$ is defined on the model output and the target $\boldsymbol{y}$. Then, based on the definition, an FL system aims to find an optimal model across all clients, which is formulated as $\min_{\boldsymbol{w} \in \mathbb{R}^d} F(\boldsymbol{w}) = \sum_{m \in [M]} p_m F_m(\boldsymbol{w})$, where $p_m = |\mathcal{D}_m|/|\mathcal{D}|$ for all $m \in [M]$, where $\mathcal{D}$ represents the entire training dataset, i.e., $\mathcal{D} = \cup_{m \in [M]} \mathcal{D}_m$. Generally, this problem can be optimized by different FL algorithms (Li et al., 2019; Karimireddy et al., 2020; Wang et al., 2020) repeating the following paradigm until convergence: 1) At the beginning of each round $t$, the server broadcasts the trainable model

parameters $\boldsymbol{w}^{(t)}$. 2) After receiving the model $\boldsymbol{w}^{(t)}$, each client $m \in [M]$ performs multi-step local updates on $\boldsymbol{w}^{(t)}$ to obtain $\boldsymbol{w}_m^{(t)}$. 3) The server collects the locally updated model parameters $\boldsymbol{w}_m^{(t)}$ from clients and aggregates them into a single global model $\boldsymbol{w}^{(t+1)}$ for next round.

This computation paradigm of FL has gained tremendous success in the past few years (Kairouz et al., 2021). However, in the context of LLM fine-tuning, the existing FL optimization algorithms encounter different challenges, which will be discussed in Section 2.1. In Section 2.2, we introduce an existing work, offsite-tuning (Xiao et al., 2023), and discuss its limitations.

## 2.1 CHALLENGES OF FEDERATED FINE-TUNING LLM

In the context of language model learning tasks, $\mathcal{M}$ can be a transformer-based model parameterized by the trainable parameters $\boldsymbol{w}$; the loss function $f$ is usually a conditional language modeling objective depending on context-target pairs $(\boldsymbol{x}, \boldsymbol{y})$, both of which are a sequence of tokens. Although FL algorithms may be employed directly for fine-tuning LLMs at first glance, there are two critical concerns in practice.

**Challenge 1: resource limitation v.s. huge model size.** As it has been shown, LLMs can exhibit remarkable emergent abilities (Zhao et al., 2023) only if their sizes are large enough. Thus, the most popular LLMs are in excessively huge sizes. For example, the sizes of LLaMA (Touvron et al., 2023) range from 7B to 65B. As one of the most popular closed-source LLMs used in many products, GPT-3 (Brown et al., 2020) is at the size of 175B parameters.

Federated fine-tuning such LLMs exerts a large burden on *local computation power* and can introduce large *communication overhead*. Kuang et al. (2023) has shown that full-parameter fine-tuning LLaMA-7B model requires 112GB GPU memory with SGD. An FL client with limited resources can hardly have applicable hardware for this size or multi-GPU training techniques. Even if a client can afford these local training costs, it was also shown that the full-parameter fine-tuning LLaMA-7B model needs to communicate with 28 GB messages, which costs 75 minutes for only uploading and downloading with 100MBps network bandwidth.

**Challenge 2: model intellectual property v.s. data privacy.** As mentioned in Section 1, although there are both open-source and closed-source LLMs, some businesses may still prefer to cooperate or purchase services from the company with closed-source LLMs. It may be because of either the desire to achieve better performance with the more well-trained models or to avoid the potential risk of commercialization in the future. However, an LLM is a critical property for a company. Therefore, it is unrealistic to expect the company to fully disclose the model parameters to any client in the collaboration. Meanwhile, existing solutions in which clients have upload their data to let the model owner conduct fine-tuning are unacceptable for some clients because of data privacy concerns or regulations.

Although a set of parameter-efficient fine-tuning (PEFT) methods, such as LoRA (Hu et al., 2021) and prompt-tuning (Lester et al., 2021; Liu et al., 2022), can be adapted to federated learning version to resolve the first challenge, they cannot resolve the second challenge. For these methods to work, the clients must access the entire pre-trained LLM, including the architecture and the model parameters, to update the PEFT components.

## 2.2 OFFSITE-TUNING IN FL: MISMATCH OF MULTI-CLIENT SETTINGS

A recent solution, offsite-tuning (Xiao et al., 2023), sheds light on resolving the abovementioned challenges. However, the proposed algorithm merely considers the setting with one model owner and one client. Suppose a pre-trained LLM has a total of $n$ layers of transformers. In the work, a repeatedly used operation is layer extraction, which extracts some layers out of the total $n$ layers of transformers to form a sub-model. We denoted this by a function $\mathsf{LayerExtract}(\mathcal{M}, L)$, which means extracting the layers with indices in $L \subseteq [n]$ from the model $\mathcal{M}$. The offsite-tuning algorithm consists of the following steps.

*Step 1: Identify the adapters in the original model.* Offsite-tuning first extracts the adapter from the original model, which is denoted as $\mathcal{A}$. In (Xiao et al., 2023; Kuang et al., 2023), $\mathcal{A}$ consists of the top and bottom two layers of transformers, namely $\mathcal{A} \leftarrow \mathsf{LayerExtract}(\mathcal{M}, L_{\mathcal{A}})$ with $L_{\mathcal{A}} =$

$\{1, 2, n - 1, n\}$. We denote $\boldsymbol{w}_{\mathcal{A}}$ as the parameters of $\mathcal{A}$. The $\boldsymbol{w}_{\mathcal{A}}$ are also the trainable parameters that can be fine-tuned by clients. The remaining part of the model is demoted as $\mathcal{E}^* = \mathcal{M} \setminus \mathcal{A}$.

*Step 2: Layer dropout to form emulator.* An emulator is extracted from the remaining part by layer dropout. The emulator is a sub-model obtained as $\mathcal{E} \leftarrow \mathsf{LayerExtract}(\mathcal{E}^*, L_{\mathcal{E}})$. Denote there are $n_{\mathcal{E}^*}$ layers transformer in $\mathcal{E}^*$. The compression ratio of the emulator is denoted as $\beta = \frac{|L_{\mathcal{E}}|}{n_{\mathcal{E}^*}}$. For convenience, we call $\mathcal{E}$ as emulator and $\mathcal{E}^*$ as non-compressed emulator. A uniform layer dropout method is used as default by Xiao et al. (2023).

*Step 3: Fine-tuning on $\mathcal{A}$ with $\mathcal{E}$.* The server sends the compressed emulator and adapter $\mathcal{E} \circ \mathcal{A}$ to a client. The client fine-tunes the $\boldsymbol{w}_{\mathcal{A}}$ of the $\mathcal{A}$ locally, keeps the $\boldsymbol{w}_{\mathcal{E}}$ frozen. Xiao et al. (2023) also claims that adding a distillation loss to align the activations between $\mathcal{E}$ and $\mathcal{E}^*$ can significantly improve the model performance.

**Inference with two combinations.** After training, we can generate two combined models, namely, Adaptor + Emulator (AdapEmu, i.e., $\mathcal{E} \circ \mathcal{A}$), and Adaptor + Full (AdapFu, i.e., $\mathcal{E}^* \circ \mathcal{A}$). As Xiao et al. (2023) describes, AdapFu performs better than AdapEmu. These two models have different functionalities in real-world scenarios: the former is adopted if the input contains sensitive information that cannot be shared with the LLM owner, e.g., drafting a petition letter, while the latter is adopted when the users aim to have better generation results, e.g., solving a math problem.

The algorithm satisfies two desiderata: 1) an adapter is fine-tuned on a client's local dataset without accessing the full model; 2) the fine-tuned adapter can not only work with the distilled emulator on the client side but also work with the full model on the server side.

**Can we do better than FedOT?** Kuang et al. (2023) migrates the offsite-tuning algorithm to the FL setting, named FedOT, as a solution to one model owner (also acting as the server in FL) with multiple clients to jointly fine-tune the trainable component. Namely, each client uploads the local version of adapter parameters $\boldsymbol{w}_{\mathcal{A}_m}^{(t)}$ to the server periodically. The server performs aggregation as $\boldsymbol{w}_{\mathcal{A}}^{(t+1)} = \sum_{m \in [M]} p_m \boldsymbol{w}_{\mathcal{A}_m}^{(t)}$ at the end of the $t$-th round, and broadcast $\boldsymbol{w}_{\mathcal{A}}^{(t+1)}$ to start the $(t+1)$-th round. As expected, FedOT can perform better than offsite-tuning individually with each client.

As shown by Xiao et al. (2023), knowledge distillation between $\mathcal{E}^*$ and $\mathcal{E}$ plays an important role in guaranteeing performance . This is because knowledge distillation (Hinton et al., 2015) can generally improve the consistency of outputs or activations between the compressed and full model on specific datasets. However, the method of knowledge distillation was not clearly specified. In the FL setting, such knowledge distillation should satisfy several prerequisites. *i) The data used for distillation must be accessible to the server.* This is because only the server can access both $\mathcal{E}$ and $\mathcal{E}^*$. *ii) The knowledge distillation loss and fine-tuning loss should be updated asynchronously.* This is because the server has no access to clients' data for fine-tuning, but the clients have no access to the non-compressed emulator for distillation.

To satisfy these prerequisites, a simple solution in FedOT is to perform knowledge distillation before sharing the emulator with the client at the beginning of FL. However, we empirically find that this approach can help little or even worsen the performance of AdapFu combination. This may be because the data distribution of the public dataset used for distillation *differs* from the data distribution of clients' datasets. Changes in input data distribution will lead to changes in the distribution of activations passing from emulators to adapter in the final layers. Thus, model distillation can only minimize the performance loss when the data follow the same distribution. However, there is no guarantee in our setting that the public dataset follows the same data distribution as the clients'. Thus, we propose our solution in the next section.

## 3    FEDBIOT: A BETTER SOLUTION TO LAYERWISE DROPOUT OF LLMS

**Overview.** Although the emulator-adapter composition seems promising for reconciling the data privacy and model intellectual property protection, we can categorize the factors of the performance loss as twofold: 1) the compressed emulator cannot completely mimic the non-compressed counterpart, and 2) the adapter has limitation of encoding the new knowledge. Thus, an ideal emulator should reproduce the counterpart of the full model as indistinguishable as possible, especially on the clients' dataset, to eliminate the first factor. A perfect emulator on the clients data distribution can

subsequently improve the the learning process of the adapter, as the training of adapter depends on the output of the emulator. Therefore, we should set our objectives from two perspectives:

• **Emulator** should be tuned towards perfectly imitating the non-compressed part in the full model, especially in extracting and encoding information on clients' datasets.

• **Adapter** should be able to digest the output of the emulator efficiently and should be encoded with the knowledge from clients' datasets effectively.

**Improvement 1.** To achieve these objectives, we first refine the choice of adapter. We propose choosing the bottom few layers[1] of transformers as the adapter. Two factors guide our choice. Regarding the computation constraints of the clients, this proposed adapter is computation-efficient because it only needs to store the activations of transformers in last few layers, leading to a lower memory cost. Furthermore, our empirical results in Section 4 verify that the LLM can acquire better performance without fine-tuning the first few layers. This phenomenon echoes with the well-known findings (Yosinski et al., 2014) in neural networks that the first few layers tend to learn general features while the last layers encode specific ones; fine-tuning LLM should focus more on processing the task-specific features.

**Improvement 2.** The next improvement proposed in this paper is on the training process. Especially, we want to ensure that the objective of emulator and adapter can be achieved simultaneously. Thus, we formulate the objectives as a bi-level optimization problem.

$$\min_{\boldsymbol{w}_{\mathcal{A}}} \sum_{m \in [M]} p_m F_m \left( \{\boldsymbol{w}_{\mathcal{A}}, \boldsymbol{w}_{\mathcal{E}}\} \right) + \frac{\epsilon}{2} \left\| \boldsymbol{w}_{\mathcal{A}} - \boldsymbol{w}_{\mathcal{A}}^{(t)} \right\|_2^2 \tag{1}$$

$$s.t. \quad \boldsymbol{w}_{\mathcal{E}} \in \arg\min_{\boldsymbol{w}_{\mathcal{E}}} \frac{1}{|\mathcal{D}_{public}|} \sum_{(\boldsymbol{x}, \boldsymbol{y}) \in \mathcal{D}_{public}} \|\mathcal{E}\left(\boldsymbol{x}; \boldsymbol{w}_{\mathcal{E}}\right) - \mathcal{E}^*\left(\boldsymbol{x}; \boldsymbol{w}_{\mathcal{E}^*}\right)\|_2^2$$
$$+ \lambda \cdot D_{KL} \left( \mathcal{M}(\boldsymbol{x}; \{\boldsymbol{w}_{\mathcal{A}}, \boldsymbol{w}_{\mathcal{E}}\}) \| \mathcal{M}(\boldsymbol{x}; \{\boldsymbol{w}_{\mathcal{A}}, \boldsymbol{w}_{\mathcal{E}^*}\})) \tag{2}$$

where $\boldsymbol{w}_{\mathcal{A}}^{(t)}$ is the adapter parameters received at the beginning of each communication round. $\mathcal{D}_{public}$ represents the public dataset on the server, which can be unlabeled. $D_{KL}(\cdot \| \cdot)$ is the KL divergence between two logits. $\epsilon$ and $\lambda$ are hyperparameters.

*The upper-level objective (Equation* (1)*).* The upper-level objective function consists of two terms. The first term represents the loss of the model on local clients' data, with the current emulator and adapter. It follows a classic weighted average loss in FL to balance the loss of different clients' heterogeneous local data. The goal of introducing this term is straightforward: by minimizing the loss of the first term, we expect the emulator-adapter combination to be improved on the local training set. The second term is a regularization of the adapter component to ensure it will be within a reasonable distance from the synchronized and broadcast adapter at the beginning of each communication round. Enforcing a restriction on the adapter's change can reduce the difference of losses for the emulator distillation after locally adapter are tuned locally on clients, so it can help the convergence of emulator distillation.

*The lower-level objective (Equation* (2)*).* The first term in the constraint is the $L_2$ difference between the activation output by the emulator and the full model. The second term is the KL divergence between the output of output distribution of the full model-adapter combination and the emulator-adapter. Although only the emulator is trainable to minimize the loss of these two terms, these two terms provide different optimization meaning for the emulator. The first term encourages the emulator to provide activations as close as possible to the full model, *excluding* the effect of the adapter. The second term ensures the emulator can provide output distributions close to the one of the full model with adapters added on.

**Federated bi-level optimization process.** We repeat the following steps for a total of $R$ rounds to solve the bi-level optimization problem. A visualization of the process is shown in Fig. 1, and the pseudocode is provided in Algo. 1.

---

[1]The bottom/last layers refer to the transformer decoders near the output, while the top/first layers refer to the part close to the input.

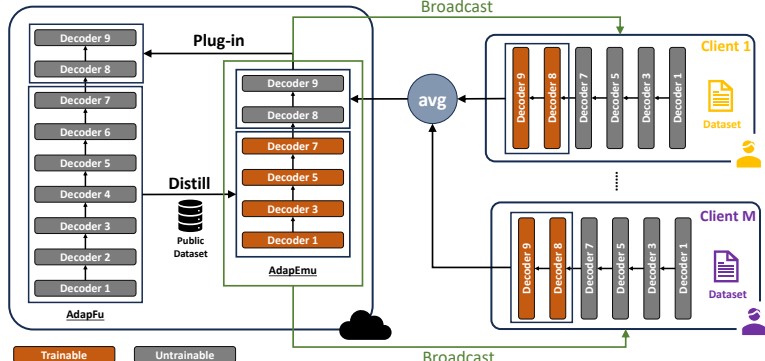

Figure 1: FedBiOT visualized with a 9-decoder transformer. Decoder $\{1, 3, 5, 7\}$ are emulator and trainable on the server only, while Decoder $\{8, 9\}$ are adapter and trainable on the clients.

*Step 1: Global model broadcasting.* The emulator is first aligned in accordance with Equation 2. Specifically, we sample a mini-batch from the public dataset and take $E$ steps to optimize Equation 2. After this, the server broadcasts the adapter $\mathcal{A}$ and the emulator $\mathcal{E}$ to the clients.

*Step 2: Local updates.* After receiving the adapter $\mathcal{A}$ and the emulator $\mathcal{E}$, the client $m$ will mark the received adapter as $\mathcal{A}^{(t)}$ and start the training. The adapter is updated for $K$ times via stochastic gradient descent (SGD) to $w_{\mathcal{A}_m}^{(t)}$, and pushed to the server.

*Step 3: Global aggregation.* The server collects all the adapters from the clients. Similar to FedOT, the server updates the global adapter via $w_{\mathcal{A}} \leftarrow \sum_{m \in [M]} p_m w_{\mathcal{A}_m}$.

**Discussion on the proposed algorithm.** The introduced algorithm can optimize the bi-level problems (i.e., Equation (1) and (2)) to an equilibrium point for both adapter and emulator. This is because when we optimize the adapter, the fixed emulator constrains its updates, and vice versa, and thereby, the emulator and adapter are distilled or trained interchangeably. At this equilibrium, the emulator can more faithfully extract and encode the information for the clients' dataset and benefit the training of adapter in reverse.

## 4 EXPERIMENTS

In this section, we conduct extensive experiments to evaluate FedBiOT on a state-of-the-art LLM across various NLP tasks. Our goal is to answer these three questions: **Q1:** What accuracy can the proposed FedBiOT achieve across different tasks in terms of two types of models, i.e., AdapEmu and AdapFu? **Q2:** How significantly does FedBiOT improve over the baseline? **Q3:** How do the terms and hyperparameters in Equation (1) and (2) affect the performance of FedBiOT?

### 4.1 EXPERIMENTAL SETUP

**Model and computation environment.** The experiments are conducted with LLaMA-7B, a pre-trained LLM proposed by Meta (Touvron et al., 2023). This model constitutes 32 decoder transformers layers with 7B parameters. The experiments are conducted on the machines with Nvidia A100 GPU cards, Intel Xeon Platinum 8369B CPU, and 512GB RAM.

**Datasets and Tasks.** In this part, we use the benchmark datasets and tasks in (Kuang et al., 2023) to train and evaluate three different NLP tasks, including code generation, math problem-solving, and question-answering. For math problem-solving, we split the GSM-8K training dataset (Cobbe et al., 2021) ensuring i.i.d. across three clients, and we assess the updated model using the GSM-8k test dataset. For code generation, we fine-tune the model with the Rosetta dataset (Chaudhary, 2023), which is partitioned based on the programming languages, and a total of nine clients separately hold the data from nine different programming languages. Regarding its evaluation, we utilize HumanEval (Chen et al., 2021), a task requiring the model to fill in the code for a given problem. For question answering, the model is trained on dolly-15K (Conover et al., 2023), which is partitioned into 8 clients based on the categories of the questions, and we evaluate the new model with the selected tasks on HELM (Liang et al., 2022). Table 4 in Appendix B gives a detailed description of

| Dropout Rate ($\beta$) | $\beta = 0.2$ | | $\beta = 0.5$ | |
|---|---|---|---|---|
| Models | AdapEmu | AdapFu | AdapEmu | AdapFu |
| Offsite-tuning | 2.27% (30/1319) | 9.78% (129/1319) | 1.97% (26/1319) | 5.61% (74/1319) |
| FedOT | 2.04% (27/1319) | 7.88% (104/1319) | 2.43% (32/1319) | 5.91% (78/1319) |
| FedBiOT (Adapter 2) | 4.32% (57/1319) | 11.14% (147/1319) | 2.27% (30/1319) | 10.16% (134/1319) |
| FedBiOT (Adapter 4) | 2.88% (38/1319) | 11.90% (157/1319) | 2.27% (30/1319) | 10.00% (132/1319) |

Table 1: Test accuracy on math problem-solving task under different dropout rates

these three tasks. As Section 3 mentions, the server will perform the emulator alignment during the model aggregation. Then, we use the Alpaca dataset (Taori et al., 2023) as the public dataset for the server to do the emulator alignment for all three NLP tasks.

**Implementation.** This work is built upon an open-source federated learning platform named FederatedScope (Xie et al., 2022). The training data are reformatted following the predesigned instructions (Chaudhary, 2023; Zhang et al., 2023).

Different from (Xiao et al., 2023; Kuang et al., 2023), we regard the last two and the last four decoders as the adapter. The experiments consider two dropout rates, i.e., $\beta \in \{0.2, 0.5\}$, and we obtain the emulators with layer dropout following Xiao et al. (2023). Without special annotation, we use the following local training setting: in each communication round, each client performs 30 local updates, and the batch size of every local update is 1. Before launching the FL training, we fine-tune the emulator for 500 iterations to generate a distilled emulator $\mathcal{E}$ towards minimizing the loss of Equation (2). During the FL training, the server takes 10 iterations to align the emulator $\mathcal{E}$ with $\mathcal{E}^*$ between two successive communication rounds after aggregating local adapters with FedAvg (Li et al., 2019). These experiments run for 1000 communication rounds, and we record the checkpoint every 100 rounds. During the training, we only fine-tune the adapter in the clients' local update procedures, and update the emulator on the server side. In other words, other parts of the pre-trained model, such as word embeddings, are frozen during the training. To avoid randomness affecting the empirical results, we take three different random seeds and report the average.

**Optimizers and Hyperparameters.** We use SGD as optimizer to solve Equation (1) and (2) on the clients (for the adapters) and the server (for the emulators), respectively. We search for the best learning rate in $\{1 \times 10^{-4}, 3 \times 10^{-4}, 5 \times 10^{-4}, 8 \times 10^{-4}, 1 \times 10^{-3}, 3 \times 10^{-3}, 5 \times 10^{-3}\}$. As for other hyperparameters related to the optimizer, we use the default setting. Furthermore, we also conduct grid search for FedBiOT-specific hyperparameters, i.e., $\epsilon$, $\lambda$, and layerwise alignment. While we demonstrate the result of the best hyperparameter combination, we discuss their influence in the ablation study in Section 4.4, and some results are recorded in Appendix C.

**Baselines.** As mentioned in Section 2.2, offsite-tuning is the only method that satisfies the constraints that fine-tuning without access to full model. Xiao et al. (2023) introduces a single-client offsite-tuning, while Kuang et al. (2023) extends it to an FL version (i.e., FedOT). We apply offsite-tuning with one single client, where all data are loaded to the client. As FedOT supports FL, we reproduce the algorithm to work on the FL tasks. In terms of the setting of the adapters and the emulators, both Offsite-tuning and FedOT treat the first two and the last two decoders as the adapter.

**Evaluation Metric.** In the experiments, we report the results on two models, i.e., AdapEmu and AdapFu, as defined in Section 2.2. The evaluation metrics for each task follow Kuang et al. (2023), and the detailed description is given in Appendix D.2.

### 4.2 QUANTITATIVE EVALUATION ON I.I.D. DATA

We demonstrate the experimental results of GSM8k provided in Table 1 and highlight the worth-noted phenomenon in the i.i.d. settings.

When we take a look at the proposed FedBiOT at different adapters' sizes, they have similar performance at the 50% dropout rate, but they perform very differently at the 20% dropout rate. To explain the results, let us identify the differences between these two settings in terms of the adapters' sizes and the non-compressed emulators' sizes: With the dropout rate of 0.2 and two layers of transformers as adapter (Adapter 2), the emulator is compressed from 30 layers (as non-compressed emulators) to 24 layers (as compressed emulator); when four layers of transformers are split as the adapter (Adapter 4), the number of layers in the emulator is reduced from 28 (as non-compressed emulators) to 22 (as compressed emulator) after dropout. Although they drop same number of layers, using 22-layer to simulate 28-layer may still be more difficult. Another intuitive explanation is the regularization part: an adapter with size 4 tend to make the adapter close to the initial setting, which is less suitable for AdapEmu updates, but it preserves the consistency between the non-compressed emulator and the emulator, making the trained adapter more suitable for AdapFu.

| Method | Model | 100 | 200 | 300 | 400 | 500 | 600 | 700 | 800 | 900 | 1000 |
|---|---|---|---|---|---|---|---|---|---|---|---|
| Offsite-tuning | AdapEmu | 2.20 | 1.22 | 1.22 | 0.61 | 1.10 | 1.71 | 1.10 | 0.73 | 1.71 | 1.22 |
| | AdapFu | 8.78 | 10.12 | 10.12 | 9.51 | 9.88 | 9.39 | 7.93 | 7.32 | 7.20 | 7.20 |
| FedOT | AdapEmu | 1.95 | 2.07 | 1.71 | 1.71 | 1.34 | 1.95 | 1.22 | 0.73 | 0.85 | 0.85 |
| | AdapFu | 9.51 | 8.17 | 7.93 | 7.32 | 6.59 | 6.59 | 7.44 | 9.88 | 10.85 | 10.85 |
| FedBiOT (Adapter 2) | AdapEmu | 4.63 | 4.88 | 5.12 | 5.00 | 4.88 | 5.24 | 5.85 | 5.12 | 4.8 | 5.61 |
| | AdapFu | 12.07 | 12.07 | 12.80 | 12.20 | 12.07 | 12.44 | 12.20 | 12.68 | 12.32 | 12.68 |
| FedBiOT (Adapter 4) | AdapEmu | 2.93 | 3.05 | 3.29 | 3.90 | 4.63 | 4.15 | 4.15 | 5.00 | 4.02 | 4.51 |
| | AdapFu | 13.05 | 13.29 | 12.80 | 12.68 | 12.80 | 12.80 | 12.80 | 12.93 | 12.44 | 12.80 |

Table 2: Pass@1 (%) in code generation task at various rounds when dropout rate is 0.2

| Method | Model | 100 | 200 | 300 | 400 | 500 | 600 | 700 | 800 | 900 | 1000 |
|---|---|---|---|---|---|---|---|---|---|---|---|
| Offsite-tuning | AdapFu | 8.66 | 8.17 | 8.29 | 7.32 | 7.07 | 6.10 | 7.07 | 7.32 | 7.68 | 8.17 |
| FedOT | AdapFu | 9.27 | 9.39 | 9.88 | 10.49 | 11.22 | 11.10 | 10.37 | 10.73 | 9.88 | 10.49 |
| FedBiOT (Adapter 2) | AdapFu | 12.44 | 11.46 | 10.85 | 11.22 | 11.71 | 10.61 | 10.61 | 10.73 | 11.46 | 11.34 |
| FedBiOT (Adapter 4) | AdapFu | 12.56 | 12.56 | 13.41 | 12.56 | 11.85 | 12.68 | 11.95 | 12.32 | 12.20 | 11.83 |

Table 3: Pass@1 (%) in code generation task at various rounds when dropout rate is 0.5. All these methods achieve 0% test accuracy with AdapEmu.

When comparing our proposed model with the baselines, we can notice a significant dominance in performance, especially in the AdapFu setting. More specifically, when the dropout rate becomes larger, the performance of AdapFu with FedBiOT decreases more mildly in contrast to other baselines. This is thanks to two factors: 1) the regularization term ensures the adapters will not change dramatically; 2) the on-the-fly distillation of the emulator with mixed losses can work better with clients' data. Although the other two baselines use a public dataset to achieve similar functionality, the deterioration may still occur due to the data domain shift and the significant information loss.

### 4.3 QUANTITATIVE EVALUATION ON NON-I.I.D. DATA

According to Table 4, code generation and question answering are two tasks split in non-i.i.d. styles. In this subsection, we elaborate the performance of our proposed FedBiOT from two perspectives, round-by-round and task-by-task, separately exemplified by these two tasks.

**Code generation.** Table 2 and 3 illustrate the best results at every 100 rounds based on different hyperparameters settings. Let us take a look at the results of the FedBiOT at different adapter sizes. It could be easily noticed that AdapEmu is generally better when the adapter size is 2, but it cannot surpass Adapter 4 regarding AdapFu. The last subsection has given an intuitive takeaway about our FedBiOT from the i.i.d. perspectives, and the non-i.i.d. settings further verify our explanation and make the phenomenon more distinct.

When comparing our proposed algorithm with the baselines, we can notice a distinct dominance, no matter at which round we stop the training. In particular, when the dropout rate is 0.2, the baselines hardly use AdapEmu to generate meaningful codes, but our proposed algorithm can complete the coding with an accuracy of up to 5.85%.

**Question Answering.** Fig. 2 and 3 (Numerical results are given in Table 10 and 11) show the evaluation results using HELM benchmark while training with Dolly-15K. Similar to previously shown results, FedBiOT (Adapter 2) has better performance than Adapter 4 in terms of AdapEmu, but worse for AdapFu. To be more specific, we can witness Adapter 2 has much better performance than Adapter 4 regarding AdapEmu in most tasks.

Although the proposed algorithm cannot surpass the baselines in all tasks regarding AdapFu, the proposed method shows outstanding performance in most cases. In general, FedBiOT provides a better averaged performance over the 6 evaluation datasets. Narrative QA is probably an out-of-distribution task to Dolly-15K because this is not a category appearing in the dataset. When the dropout rate gets larger, FedBiOT with AdapFu improves because most information/knowledge is still preserved on the raw model, while fine-tuning on the adapter will not have a significant impact on weakening the model's lack of generalization ability.

### 4.4 ABLATION STUDY

Compared to two baselines, we introduce two more terms, i.e., the regularization term for the adapter optimization on the client side, and the distillation term for the emulator optimization on the server side. In addition, the implementation of FedBiOT follows a bi-level optimization, and therefore, it is non-trivial to answer how many updates the server should take to update the emulator between two successive communication rounds. Furthermore, although we take 1000 communication rounds to fine-tune, the test accuracy may not keep increasing as the traditional FL algorithms do. In other

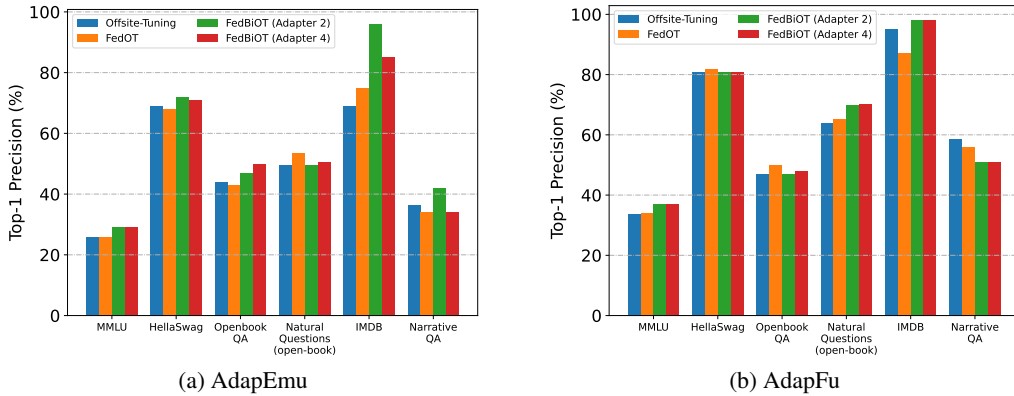

(a) AdapEmu         (b) AdapFu

Figure 2: Test accuracy in six types of question-answering task when dropout rate is 0.2

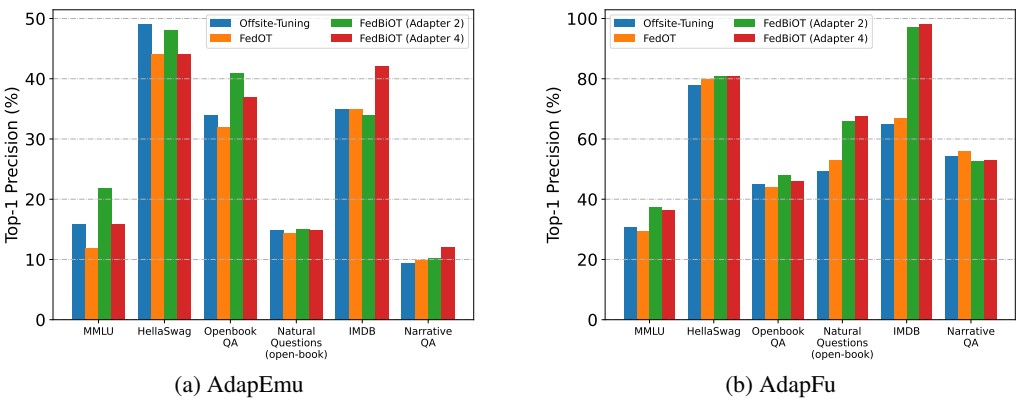

(a) AdapEmu         (b) AdapFu

Figure 3: Test accuracy in six types of question-answering task when dropout rate is 0.5

words, the best performance probably appears during the training process. We conduct the ablation study to explore the answers of these questions and verify the necessity of the new components. Due to the space limite, we defer the details to Appendix C but summarize some key findings.

• The layerwise alignment on the emulator is not necessary for our proposed FedBiOT because the results show no performance improvement compared to the alignment between the outputs of the non-compressed emulator and the emulator.

• With $\lambda$ getting larger, a significant improvement can be witnessed for AdapEmu. But for AdapFu, the performance depends on the compression ratio: larger $\lambda$ can improve performance with a larger dropout rate (e.g., 0.5), but worsen with a smaller dropout rate (e.g., 0.2).

• The regularization term benefits the training of AdapEmu and AdapFu because both models' test accuracies are boosted by comparing $\epsilon = 0$ and $\epsilon \neq 0$.

• It is hard to say how many times emulator updates can bring the best performance. When we set it to 10, the performance will likely stay in the middle range. For the communication rounds, the checkpoints at the 500th round are worth evaluating because incredible results are observed.

## 5 CONCLUSION

In this paper, we introduce FedBiOT, a federated learning strategy that protects the intellectual property of LLMs while enabling a large model fine-tuning on the clients. By splitting the LLM into two parts, namely adapters and non-compressed emulators, we compress the non-compressed emulators as emulators via layer-wise dropout. The proposed algorithm follows a bi-level optimization process: The clients receive the adapter and the emulator and make updates on the adapters, and the server aggregates the adapters from the clients and fine-tunes emulators to guarantee the alignment between compressed and non-compressed emulators. Extensive experiments show the superiority of the proposed FedBiOT working with LLaMA, where it can numerically achieve more than 4% accuracy improvement than the existing baselines (i.e., Offsite-tuning and FedOT) in all tasks (i.e., math problem solving, code generation, and question answering).

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

## A ALGORITHM DESCRIPTION

---

**Algorithm 1** FedBiOT
***

**Input:** learning rate $\eta$, local updates $K$, global model alignment steps $E$, strength of full model alignment $\lambda$, local update regularization $\epsilon$, total communication rounds $R$, pre-trained LLM $\mathcal{M}$ with parameter $\boldsymbol{w}$, adapter size $s$, compression ratio $\beta$, number of clients $M$.

1: $\boldsymbol{w}_{\mathcal{A}}, \boldsymbol{w}_{\mathcal{E}}, \boldsymbol{w}_{\mathcal{E}^*} \leftarrow \mathsf{LayerExtract}(\mathcal{M}, s, \beta)$           ▷ See Algo. 2 for details
2: **for** $t = 0, \ldots, R - 1$ **do**
3:      **for** $e = 0, \ldots, E - 1$ **do**
4:          Randomly sample $(\boldsymbol{x}, \boldsymbol{y})$ from the public dataset $\mathcal{D}_{public}$
5:          Optimize $\boldsymbol{w}_{\mathcal{E}}$ via Eq. (2) with the given sample $(\boldsymbol{x}, \boldsymbol{y})$
6:      **end for**
7:      Communicate the model $\{\boldsymbol{w}_{\mathcal{A}}, \boldsymbol{w}_{\mathcal{E}}\}$ with the clients $m \in [M]$
8:      **for** $m \in [M]$ **in parallel do**
9:          Initialize $\boldsymbol{w}_{\mathcal{A}}^{(t)} \leftarrow \boldsymbol{w}_{\mathcal{A}}, \boldsymbol{w}_{\mathcal{A},m} \leftarrow \boldsymbol{w}_{\mathcal{A}}$
10:          **for** $k = 0, \ldots, K - 1$ **do**
11:              $\boldsymbol{w}_{\mathcal{A},m} \leftarrow \boldsymbol{w}_{\mathcal{A},m} - \eta \left( \nabla_{\boldsymbol{w}_{\mathcal{A},m}} F(\{\boldsymbol{w}_{\mathcal{A},m}, \boldsymbol{w}_{\mathcal{E}}\}) + \epsilon \left( \boldsymbol{w}_{\mathcal{A},m} - \boldsymbol{w}_{\mathcal{A}}^{(t)} \right) \right)$
12:          **end for**
13:          Communicate $\boldsymbol{w}_{\mathcal{A},m}$ with the server
14:      **end for**
15:      $\boldsymbol{w}_{\mathcal{A}} \leftarrow \frac{1}{M} \sum_{m \in [M]} \boldsymbol{w}_{\mathcal{A},m}$
16: **end for**
17: **return** AdapEmu $\{\boldsymbol{w}_{\mathcal{A}}, \boldsymbol{w}_{\mathcal{E}}\}$, AdapFu $\{\boldsymbol{w}_{\mathcal{A}}, \boldsymbol{w}_{\mathcal{E}^*}\}$

---

**Algorithm 2** LayerExtract
***

**Input:** pre-trained LLM $\mathcal{M}$ (layer index starts from 0), adapter size $s$, compression ratio $\beta$.

1: Get the size of model: $n \leftarrow |\mathcal{M}|$
2: Compute the number of layers in the compressed model $n' \leftarrow \lfloor \beta(n - s) \rfloor$
3: Initialize $\mathcal{E}^* \leftarrow \{\mathcal{M}_0, \ldots \mathcal{M}_{n-s-1}\}$, adapter $\mathcal{A} \leftarrow \{\mathcal{M}_{n-s}, \ldots, \mathcal{M}_{n-1}\}$
4: Initialize emulator $\mathcal{E} \leftarrow \{\}$
5: Compute $stride \leftarrow (n - s - 1)/(n' - 1)$
6: **for** $j = 0, \ldots, n' - 1$ **do**
7:      Append $\mathcal{M}_{\lfloor j \times stride \rfloor}$ to emulator, i.e., $\mathcal{E} \leftarrow \mathcal{E} + \{\mathcal{M}_{\lfloor j \times stride \rfloor}\}$
8: **end for**
9: **return** $\boldsymbol{w}_{\mathcal{A}}, \boldsymbol{w}_{\mathcal{E}}, \boldsymbol{w}_{\mathcal{E}^*}$

---

## B   DATASET DETAILS

| Task | Training Dataset | # training samples | # clients | Partition Rules | Max. | Min. | Std. | Test Dataset | # test samples |
|------|------------------|--------------------|-----------|------------------|------|------|------|--------------|----------------|
| Math Problem Solving | GSM-8K | 7473 | 3 | i.i.d. | 2491 | 2491 | 0 | GSM-8K | 1319 |
| Code Generation | Rosetta | 7954 | 9 | Prog. Lang. | 1172 | 439 | 236.94 | HumanEval | 164 |
| Question Answering | Dolly | 15015 | 8 | Category | 3611 | 711 | 795.06 | Helm | NA |

Table 4: Dataset details for LLM training and evaluation

## C   ABLATION STUDY

| Dropout Rate | 0.2 | | 0.5 | |
|--------------|---------|--------|---------|--------|
| Models | AdapEmu | AdapFu | AdapEmu | AdapFu |
| $\lambda = 0$ | 2.88% | 10.92% | 2.27% | 8.61% |
| $\lambda = 0.1$ | 3.26% | 10.54% | 2.27% | 8.95% |
| $\lambda = 0.25$ | 3.94% | 10.84% | 1.82% | 8.72% |
| $\lambda = 0.5$ | 4.32% | 10.16% | 2.12% | 8.95% |
| $\lambda = 1.0$ | 4.25% | 10.16% | 2.20% | 10.16% |

Table 5: Comparison with different $\lambda$s in GSM8k with Adapter 2 without layerwise distillation

| Layerwise | ✗ | | ✓ | |
|-----------|---------|--------|---------|--------|
| Models | AdapEmu | AdapFu | AdapEmu | AdapFu |
| $\lambda = 0$ | 2.88% | 10.92% | 3.18% | 9.78% |
| $\lambda = 0.25$ | 3.94% | 10.84% | 3.41% | 9.78% |
| $\lambda = 0.5$ | 4.32% | 10.16% | 3.71% | 10.00% |
| $\lambda = 1.0$ | 4.25% | 10.16% | 3.48% | 10.46% |

Table 6: Comparison on the effect of layerwise alignment in GSM8k with Adapter 2 with dropout rate 0.2

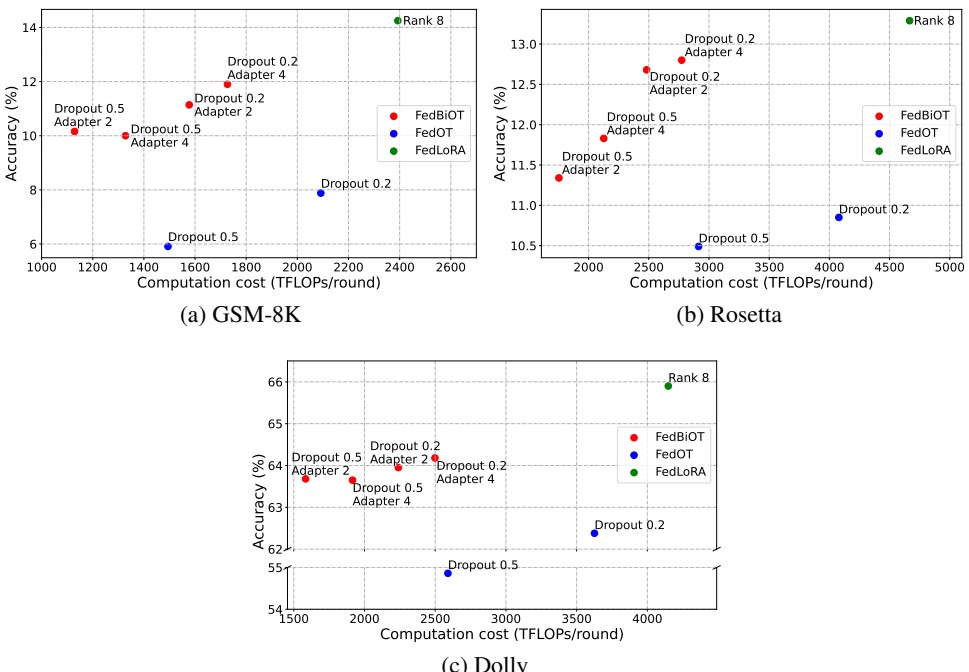

(a) GSM-8K

(b) Rosetta

(c) Dolly

Figure 4: Computation per round v.s. the test accuracy after 1000 rounds of fine-tuning (Zoom in for the best view)

| $\lambda$ | 0.0 | | 1.0 | |
|---|---|---|---|---|
| Models | AdapEmu | AdapFu | AdapEmu | AdapFu |
| $\epsilon = 0.0$ | 2.88% | 10.24% | 4.25% | 9.93% |
| $\epsilon = 0.25$ | 3.03% | 10.16% | 4.02% | 10.16% |
| $\epsilon = 0.5$ | 2.88% | 10.92% | 4.02% | 10.08% |
| $\epsilon = 1.0$ | 3.03% | 10.99% | 4.25% | 10.16% |

Table 7: Comparison on the effect of $\epsilon$ in GSM8k with Adapter 2 with dropout rate 0.2

| $\lambda$ | 0.0 | | 1.0 | |
|---|---|---|---|---|
| Models | AdapEmu | AdapFu | AdapEmu | AdapFu |
| $E = 5$ | 3.03% | 10.61% | 4.25% | 10.08% |
| $E = 10$ | 3.03% | 10.99% | 4.25% | 10.16% |
| $E = 30$ | 3.41% | 9.48% | 4.47% | 10.77% |
| $E = 50$ | 3.18% | 9.63% | 4.40% | 10.54% |

Table 8: Comparison on the effect of alignment steps in GSM8k with Adapter 2 with dropout rate 0.2

| $\lambda$ | 0.0 | | 1.0 | |
|---|---|---|---|---|
| Models | AdapEmu | AdapFu | AdapEmu | AdapFu |
| $R = 200$ | 3.41% | 9.70% | 3.71% | 9.47% |
| $R = 500$ | 3.94% | 9.48% | 4.09% | 11.14% |
| $R = 800$ | 3.11% | 9.55% | 4.62% | 10.69% |
| $R = 1000$ | 3.03% | 10.99% | 4.25% | 10.16% |

Table 9: Comparison on the effect of the number of communication rounds $R$ alignment in GSM8k with Adapter 2 with dropout rate 0.2

| Methods | Models | MMLU | HellaSwag | OpenbookQA | NaturalQuestions (open-book) | IMDB | NarrativeQA | Average |
|---|---|---|---|---|---|---|---|---|
| Offsite-Tuning | AdapEmu | 25.9 | 69 | 44 | 49.6 | 69 | 36.4 | 48.98 |
| | AdapFu | 33.7 | 81 | 47 | 63.9 | 95 | **58.7** | 63.21 |
| FedOT | AdapEmu | 25.8 | 68 | 43 | *53.4* | 75 | 34.2 | 49.90 |
| | AdapFu | 34.1 | **82** | **50** | 65.3 | 87 | 55.9 | 62.38 |
| FedBiOT (Adapter 2) | AdapEmu | 29.0 | *72* | 47 | 49.5 | *96* | *42.0* | 55.92 |
| | AdapFu | **37.0** | 81 | 47 | 69.8 | 98 | 50.9 | 63.95 |
| FedBiOT (Adapter 4) | AdapEmu | *29.3* | 71 | *50* | 50.7 | 85 | 34.2 | 53.37 |
| | AdapFu | **37.0** | 81 | 48 | **70.2** | **98** | 50.9 | **64.18** |

Table 10: Test accuracy on HELM while training using Dolly when dropout rate is fixed to 0.2

| Methods | Models | MMLU | HellaSwag | OpenbookQA | NaturalQuestions (open-book) | IMDB | NarrativeQA | Average |
|---|---|---|---|---|---|---|---|---|
| Offsite-Tuning | AdapEmu | 15.9 | *49* | 34 | 14.9 | 35 | 9.4 | 26.37 |
| | AdapFu | 30.6 | 78 | 45 | 49.2 | 65 | 54.2 | 53.67 |
| FedOT | AdapEmu | 11.9 | 44 | 32 | 14.4 | 35 | 9.9 | 24.53 |
| | AdapFu | 29.4 | 80 | 44 | 52.9 | 67 | **55.9** | 54.86 |
| FedBiOT (Adapter 2) | AdapEmu | *21.8* | 48 | *41* | *15.1* | 34 | 10.2 | *28.35* |
| | AdapFu | **37.5** | **81** | **48** | 65.9 | 97 | 52.7 | **63.68** |
| FedBiOT (Adapter 4) | AdapEmu | 15.9 | 44 | 37 | 14.8 | *42* | *12.0* | 27.62 |
| | AdapFu | 36.2 | **81** | 46 | **67.7** | **98** | 53.0 | 63.65 |

Table 11: Test accuracy on HELM while training using Dolly when dropout rate is fixed to 0.5

## D   PROMPT FOR TRAINING AND EVALUATION

Recent studies (Ouyang et al., 2022; Wang et al., 2022; Wei et al., 2021; Zhang et al., 2023) have demonstrated that prompted inputs can improve an LLM's training and evaluation performance when compared to raw inputs. As there are some existing works (Zhang et al., 2023; Taori et al., 2023; Wei et al., 2022) providing a strategy about how to train and evaluate a LLaMA, we follow these works and introduce how we design the prompt and reformat the inputs for different training and testing tasks.

### D.1   TRAINING DATASETS

**Dolly.**   The training samples of the question-answering dataset are labeled by "instruction" (i.e., question), "context" (i.e., a passage serves as a reference to the question, which can be empty/none), "response" (i.e., the answer to the question), and "category" (i.e., the class of the question, including creative writing, general QA, closed QA, open QA, summarization, information extraction, classification, brainstorming). In specific, we partition the dataset based on the samples' categories and ensure that eight clients hold different categories. Besides, we construct the input using the following templates, which is determined if the "context" is empty. Last but not least, "response" is the ground truth of each training sample.

| "context" is None | Below is an instruction that describes a task, paired with an input that provides further context. Write a response that appropriately completes the request. 

 ### Instruction: 
 {instruction} 

 ### Response: |
|---|---|
| "context" is not None | Below is an instruction that describes a task, paired with an input that provides further context. Write a response that appropriately completes the request. 

 ### Instruction: 
 {instruction} 

 ### Input: 
 {context} 

 ### Response: |

**GSM-8K.**   As a math problem-solving dataset, GSM-8K consists of the training samples labeled by "question" and "answer". Then, we reformat the question with the below instruction as the input and treat the answer as the ground truth.

| Below is an instruction that describes a task. Write a response that appropriately completes the request. 

 ### Instruction: 
 {question} 

 ### Response: |
|---|

**Rosetta.**   Each training sample is comprised of three parts: "instruction" (i.e., the description of a coding task), "input" (i.e., the required programming language), and "output" (i.e., the sampled code for the given task). Therefore, we reformat "instruction" and "input" of a sample with the below template and take "output" as the ground truth. We disjoint the dataset across nine clients where each client holds a programming language only.

Below is an instruction that describes a task, paired with an input that provides further context. Write a response that appropriately completes the request.

### Instruction:
{instruction}

### Input:
{input}

### Response:

**Alpaca.** This dataset serves as a public dataset and covers a wide range of tasks. Specifically, each training sample includes "instruction", "input", and "output". We construct the training data using the following templates determined by "input".

| "input" is None | Below is an instruction that describes a task, paired with an input that provides further context. Write a response that appropriately completes the request.

### Instruction:
{instruction}

### Response: |
| --- | --- |
| "input" is not None | Below is an instruction that describes a task, paired with an input that provides further context. Write a response that appropriately completes the request.

### Instruction:
{instruction}

### Input:
{input}

### Response: |

## D.2 TESTING DATASETS AND EVALUATION

**HELM.** HELM (Liang et al., 2022) is a benchmark that contains a wide range of NLP tasks. We upload the well-trained models to the benchmark and evaluate them on six selected tasks (i.e., MMLU, HellaSwag, Openbook QA, Natural Questions (open-book), IMDB, and Narrative QA). In this paper, we report the results of the exact match.

**HumanEval.** This is a task for Python code autofill, which consists of 164 test samples. Each test sample is constituted with "task id", "prompt" (i.e., Task description with partial codes), "entry point" (i.e., the function to be achieved), "canonical solution" (i.e., a sampled solution), and "test" (i.e., evaluate if the generated code can obtain the correct answer based on the given input). In this task, we use "prompt" as the input and generate five versions of codes using a given model. We compile the code and check if it can pass the given "test". Let $c$ be the number of correct codes generated by LLM and passed unit tests, and therefore, Pass@1 can be computed via

$$\text{Pass@1} = \mathbb{E}_{\text{problems}}\left(c/5\right)$$

**GSM-8K.** Similar to its training dataset, GSM-8K consists of "question" and "answer". Different from the training, we offer eight demos before the math problem (a.k.a. eight-shot learning; see the following prompt for more details). Therefore, the model will generate a chain-of-thought solution, and we will assess if it can exactly match the final correct answer.

Below is a set of demos followed by an instruction that describes a task. Write a response that appropriately completes the request.

### Instruction:
There are 15 trees in the grove. Grove workers will plant trees in the grove today. After they are done, there will be 21 trees. How many trees did the grove workers plant today?

### Response:
There are 15 trees originally. Then there were 21 trees after some more were planted. So there must have been 21 - 15 = 6. The answer is 6.

### Instruction:
If there are 3 cars in the parking lot and 2 more cars arrive, how many cars are in the parking lot?

### Response:
There are originally 3 cars. 2 more cars arrive. 3 + 2 = 5. The answer is 5.

### Instruction:
Leah had 32 chocolates and her sister had 42. If they ate 35, how many pieces do they have left in total?

### Response:
Originally, Leah had 32 chocolates. Her sister had 42. So in total they had 32 + 42 = 74. After eating 35, they had 74 - 35 = 39. The answer is 39.

### Instruction:
Jason had 20 lollipops. He gave Denny some lollipops. Now Jason has 12 lollipops. How many lollipops did Jason give to Denny?

### Response:
Jason started with 20 lollipops. Then he had 12 after giving some to Denny. So he gave Denny 20 - 12 = 8. The answer is 8.

### Instruction:
Shawn has five toys. For Christmas, he got two toys each from his mom and dad. How many toys does he have now?

### Response:
Shawn started with 5 toys. If he got 2 toys each from his mom and dad, then that is 4 more toys. 5 + 4 = 9. The answer is 9.

### Instruction:
There were nine computers in the server room. Five more computers were installed each day, from monday to thursday. How many computers are now in the server room?

### Response:
There were originally 9 computers. For each of 4 days, 5 more computers were added. So 5 * 4 = 20 computers were added. 9 + 20 is 29. The answer is 29.

### Instruction:
Michael had 58 golf balls. On tuesday, he lost 23 golf balls. On wednesday, he lost 2 more. How many golf balls did he have at the end of wednesday?

### Response:
Michael started with 58 golf balls. After losing 23 on tuesday, he had 58 - 23 = 35. After losing 2 more, he had 35 - 2 = 33 golf balls. The answer is 33.

### Instruction:
Olivia has $23. She bought five bagels for $3 each. How much money does she have left?

### Response:
Olivia had 23 dollars. 5 bagels for 3 dollars each will be 5 x 3 = 15 dollars. So she has 23 - 15 dollars left. 23 - 15 is 8. The answer is 8.

### Instruction:
{question}

### Response:

