# OpenReview forum: "FedBiOT: a solution for federated large language model fine-tuning with intellectual property protection"
_ICLR.cc/2024/Conference — Submitted to ICLR 2024_

### Official Review · Reviewer_nUtE · 2023-10-22

**Soundness:** 2 fair
**Presentation:** 3 good
**Contribution:** 2 fair
**Rating:** 5
**Confidence:** 3

**Summary:**

Due to privacy concerns, data owners and large language model (LLM) owners are reluctant to share data and models. This paper proposes FedBiOT, a method that ensures data privacy while enabling fine-tuning of LLMs on federated learning tasks. It formulates and solves a bi-level optimization problem to distill an emulator on a public dataset that can support local fine-tuning on private datasets without disclosing the LLM.

**Strengths:**

1. The combination of LLM and federated learning is interesting.
2. The problem formulation is well presented.
3. The layer selection and dropout mechanism is interesting.

**Weaknesses:**

However, there are some improvements for the paper:
1. The number of clients is very small. In section 4.1, the number of clients is 4, which is relatively very small compared with that in real FL settings.
2. The idea is straightforward, which is presented in existing works, e.g., Yosinski et al., 2014.
3. The selection of dropout rate is not well elaborated.
4. Tables 1 and 2 are not clear. The first line is not explained. The unit can be added, and the meaning of the numbers can be explained.
5. The experimentation show that the performance of FedBiOT may be inferior than baselines.
6. The classic FL approaches can be added as baselines.

**Questions:**

1. Tables 1 and 2 are not clear. The first line is not explained. What is the unit can be added?
2. The classic FL approaches can be added as baselines. I wonder if the authors can compare FedBiOT with classic approaches.

---

> ### Author Response · Authors · 2023-11-18
>
> We are grateful for your careful review and constructive comments. Based on your comments, we would like to make the following clarification.
>
> > **[W1]** The number of clients is very small. In section 4.1, the number of clients is 4, which is relatively very small compared with that in real FL settings.
>
> Thanks for your question. There are two settings using more than four clients, i.e., 8 for Dolly (Question Answering) and 9 for Rosetta (Code Generation). Our experiments follow a well-developed benchmark, i.e., FederatedScope LLM [1]. Also, we notice some state-of-the-art works [2, 3, 4] discussing LLM in FL following the same setting as well.
>
> **References:**
>
> [1] FederatedScope-LLM: A Comprehensive Package for Fine-tuning Large Language Models in Federated Learning
>
> [2] Towards Building the Federated GPT: Federated Instruction Tuning
>
> [3] Federated Tuning for Black Box Large Models
>
> [4] FedBPT: Efficient Federated Black-box Prompt Tuning for Large Language Models
>
>
> > **[W2]** The idea is straightforward, which is presented in existing works, e.g., Yosinski et al., 2014.
>
> Thanks for your comments. We point out the differences that our idea has been done by Yosinski et al. (2014) because it does not introduce an approach to fine-tune a deep model. Instead, Yosinski et al. (2014) discovered that the first few layers of a deep model are used to learn general features while the rest encode specific features. Motivated by this discovery, we propose an algorithm where the clients fine-tune the last few layers rather than the entire model because our work focuses more on processing task-specific features.
>
> > **[W3]** The selection of dropout rate is not well elaborated.
>
> Thanks for your question. Removing fewer layers will lead to significant intellectual property concerns while removing more layers will lead to poor performance. To find the trade-off between these two aspects, we discuss the setting of 0.2 and 0.5.
>
> There are 32 decoders in a LLaMA-7B model. If the dropout rate is 0.2 and the adapter size is 2, then the emulator consists of 24 decoders, and the total size of the compressed model is 26. Similarly, the sizes under different adapter size and dropout rate settings are given in the following table:
>
> |Dropout rate|Adapter size|Emulator size|Total size|
> |:--------:|:-------:|:-------:|:------:|
> |0.2|2|24|26|
> |0.2|4|22|26|
> |0.5|2|15|17|
> |0.5|4|14|18|
>
> > **[W4]** Tables 1 and 2 are not clear. The first line is not explained. The unit can be added, and the meaning of the numbers can be explained.
>
> Thanks for your question. 0.2 means we remove 20% of layers, while 50% means we remove 50% of layers (as mentioned in **[W3]**). We will make it clear in the caption. In Table 2/3, the numbers in the first row refer to the indices of the communication rounds.
>
> > **[W5]** The experimentation show that the performance of FedBiOT may be inferior than baselines.
>
> Thanks for your question. We have noticed that FedBiOT may be inferior to baselines and explained why it happens in experiments. We again highlight those fall-behinds and reiterate the reasons:
>
> - **Code Generation (Table 2/3).** We capture the checkpoints every 100 communication rounds. Generally speaking, FedBiOT performs better. However, since the results are reported at certain checkpoints, FedBiOT may suffer from a sudden drop such that other baselines surpass our proposed method.
> - **Question Answering (Table 4/5; now Figure 2/3 in revision).** The proposed method constantly outperforms other baselines except for NarrativeQA. In detail, AdapEmu in FedBiOT outperforms that in other baselines, but AdapFu in FedBiOT has poor performance. Possibly, the emulator cannot imitate the raw model in terms of NarrativeQA because the public dataset does not possess the relevant data. As a result, even if the adapter can answer NarrativeQA based on the given emulator, the ability cannot plug in the full model.
>
> > **[W6]** The classic FL approaches can be added as baselines.
>
> Thanks for your suggestion. The classic FL approaches will utilize the entire full model for fine-tuning, which conflicts with our setting to protect intellectual properties. We add these results in Figure 4, where our proposed work utilizes much fewer computation costs and achieves the comparable performance of FedLoRA, a classic FL approach fine-tuning LLM.
>
> **Appendix:** The result shows FedLoRA, and we fix the rank to be 8 and try the scaling factor $\alpha=32, 64, 128$
>
> - GSM8k:
>
> |Method|Accuracy|
> |---|---|
> |Full Model (LoRA)|14.25% (188/1319)|
>
> - Rosetta:
>
> |Method|Accuracy|
> |---|---|
> |Full Model (LoRA)|13.29%|
>
> - Dolly:
>
> |Method|MMLU|HellaSwag|OpenbookQA|Natural Questions (open-book)|IMDB|Narrative QA|
> |---|:-:|:-:|:-:|:-:|:-:|:-:|
> |Full Model (LoRA)| 36.5 | 82 | 52 | 71.3 | 98 | 55.6 |

---

### Official Review · Reviewer_hDuF · 2023-10-28

**Soundness:** 3 good
**Presentation:** 3 good
**Contribution:** 2 fair
**Rating:** 5
**Confidence:** 4

**Summary:**

This paper considers a relatively new setting: federated learning over large language models. There are two key considerations of this paper: limited computation resource of clients and intellectual property of server's full LLM. Based on these motivations, this paper proposes an FL algorithm FedBiOT, which trains adapter and emulator in a bi-level optimization manner. Experiments on three datasets shoe the effectiveness of FedBiOT by comparing with two baselines.

**Strengths:**

- This paper considers a relatively new setting in federated learning and large language models.
- This paper proposes a new FL algorithm FedBiOT, which trains adapter over emulator to achieve parameter-efficient tuning.
- Experiments show the effectiveness of FedBiOT by comparing with two baselines.

**Weaknesses:**

- The contributions need to be clarified. For me, I think the topic of this paper is interesting and worth exploring. However, it is not so clear what are the main contributions of this paper since previous work [1] has considered such setting and proposed FedOT (federated learning with offsite-tuning). Are the main contributions lying on improving FedOT via a bi-level optimization approach?
- The motivations need to be further clarified. This paper claims that the clients cannot obtain the full model due to intellectual property of LLM. However, I wonder if such claim still holds after the release of Llama2.
- Some meaningful experiments are missing.
  - Some experiments for reference. It would be more helpful if the authors can provide the results when clients can obtain the full model, such that we could see how large the gap is.
  - Computation resources comparisons. This method requires more training resources (e.g., more training steps) compared to baselines. However, this paper does not show such comparisons, which would promote readers' understanding.
- Some confusions:
  - "Improvement 1" at page 5. What are the definations of bottom / first / last layers. Suggest consistent expressions like first / last.

Currently, my rating is between 5 and 6. I would consider re-rating if the authors can address the above concerns.

[1] Weirui Kuang, Bingchen Qian, Zitao Li, Daoyuan Chen, Dawei Gao, Xuchen Pan, Yuexiang Xie, Yaliang Li, Bolin Ding, and Jingren Zhou. Federatedscope-llm: A comprehensive package for fine-tuning large language models in federated learning.

**Questions:**

- According to this sentence "We apply offsite-tuning with one single client, where all data are loaded to the client.", it seems like the Offsite-tuning is training over the gathered dataset of all clients. But why its performance is quite low? Please describe how you implement offsite-tuning with more details.

---

> ### Author Response · Authors · 2023-11-18
>
> We appreciate your insightful comments on our work. Based on your comments, we would like to clarify your concerns as follows.
>
> ***
>
> > **[W1]** The contributions need to be clarified. For me, I think the topic of this paper is interesting and worth exploring. However, it is not so clear what are the main contributions of this paper since previous work [1] has considered such setting and proposed FedOT (federated learning with offsite-tuning). Are the main contributions lying on improving FedOT via a bi-level optimization approach?
>
> Thank you for your comments. First, let us explain the relationship among Offsite Tuning, FedOT, and FedBiOT (our proposed work):
>
> - Offsite Tuning is the first work that points out the bottleneck of LLM fine-tuning, i.e., the LLM owner cannot transmit the model to the clients, while the data holder cannot share the data with the server.
> - FedOT considers there are multiple data holders for a specific task and extends the proposed offsite tuning algorithm to a federated learning setting.
> - FedBiOT (ours) highlights their expensive computation costs and explicit model and data privacy risks. To address these challenges, we introduce a more efficient bi-level optimization approach, FedBiOT, that guarantees intellectual properties.
>
> > **[W2]** The motivations need to be further clarified. This paper claims that the clients cannot obtain the full model due to intellectual property of LLM. However, I wonder if such claim still holds after the release of Llama2.
>
> Thanks for your question. Looking back at the development of GPT, we know that OpenAI released the checkpoints of GPT-2 and GPT-3 but kept mysterious for GPT-3.5 and GPT-4. Thus, there is no guarantee that those companies will still be willing to open-source more powerful models in the future. Our proposed technique can be an alternative besides the binary open-source v.s. closed-source options.
>
> As a resilient and flexible solution, the proposed method can build a win-win community: The LLM holders release a compressed model, which the clients train with their local data. The server can aggregate the updated models from the clients and utilize them to enhance the model performance on the downstream task.
>
> > **[W3]** Some meaningful experiments are missing.
> > - Some experiments for reference. It would be more helpful if the authors can provide the results when clients can obtain the full model, such that we could see how large the gap is.
> > - Computation resources comparisons. This method requires more training resources (e.g., more training steps) compared to baselines. However, this paper does not show such comparisons, which would promote readers' understanding.
>
> Thank you for your suggestions. We provide you with concrete results when fine-tuning with the full model as follows. (Note: The result here shows LoRA, and we fix the rank to be 8 and try the scaling factor $\alpha=32, 64, 128$)
>
> - GSM8k:
>
> |Method|Accuracy|
> |---|---|
> |Full Model (LoRA)|14.25% (188/1319)|
>
> - Rosetta:
>
> |Method|Accuracy|
> |---|---|
> |Full Model (LoRA)|13.29%|
>
> - Dolly:
>
> |Method|MMLU|HellaSwag|OpenbookQA|Natural Questions (open-book)|IMDB|Narrative QA|
> |---|:-:|:-:|:-:|:-:|:-:|:-:|
> |Full Model (LoRA)| 36.5 | 82 | 52 | 71.3 | 98 | 55.6 |
>
>
> For computation comparison, We prepare a scatter figure in the appendix to show the difference in terms of the calculation in Figure 4. As we can see, the full model consumes a large amount of computation effort.
>
> > Some confusions:
> > - "Improvement 1" at page 5. What are the definations of bottom / first / last layers. Suggest consistent expressions like first / last.
>
> Thanks for your suggestion. We now add a footnote to clarify these three words. In detail, The bottom/last layers refer to the transformer decoders near the output, while the top/first layers refer to the part close to the input.

---

> > ### Author Response · Authors · 2023-11-18
> >
> > > **[Q1]** According to this sentence "We apply offsite-tuning with one single client, where all data are loaded to the client.", it seems like the Offsite-tuning is training over the gathered dataset of all clients. But why its performance is quite low? Please describe how you implement offsite-tuning with more details.
> >
> > Thanks for your question. As Kuang et al. [1] have implemented Offsite-Tuning [2] in their FederatedScope framework, we utilize their codes and report the results of Offsite-Tuning on our designated tasks. In detail, they reproduce the algorithm with the following steps:
> > - Step 1: Choose the adapter and the emulator from the full model and form a compressed model. In detail, the adapter is the first two and the last two decoders.
> > - Step 2: Use the public dataset to align the knowledge between the full model and the compressed model while fixing the adapter
> > - Step 3: Fine-tune the adapter until the model converges
> >
> > We owe its poor performance to two reasons. First, the emulator is yet to correctly align with the raw model. As offsite-tuning ignores the effect of the final two decoders (i.e., the latter part of the adapter), they can lead to a significant discrepancy even if their embeddings at the middle layers are very similar. Second, since the emulator never updates during the fine-tuning stage, the emulator fails to imitate the functionality of the raw model as the training rounds fly. Therefore, training the adapter of AdapEmu cannot boost the performance of the plug-in full model (i.e., AdapFu).
> >
> > **Reference:**
> >
> > [1] FederatedScope-LLM: A Comprehensive Package for Fine-tuning Large Language Models in Federated Learning
> >
> > [2] Offsite-Tuning: Transfer Learning without Full Model

---

### Official Review · Reviewer_4sKJ · 2023-11-01

**Soundness:** 2 fair
**Presentation:** 3 good
**Contribution:** 2 fair
**Rating:** 6
**Confidence:** 1

**Summary:**

This paper introduces FedBiOT, a method that guarantees the clients’ data privacy and avoids the disclosure of an LLM. The authors conduct extensive experiments on LLaMA-7B training for various federated learning tasks and witness significant improvements over existing baselines.

**Strengths:**

The topic is timely and interesting.

**Weaknesses:**

1. The experimental evaluation was only implemented in LLaMA-7B. How does it work on other mainstream models such as ChatGPT2?
2. In the experiment, federated learning only considered 8 clients. There is a lack of experiments that vary the number of clients and the number of training samples each client own.

**Questions:**

See above.

---

> ### Author Response · Authors · 2023-11-18
>
> Thanks a lot for providing us with valuable and constructive feedback. Based on your comments, we would like to address your concerns below.
>
> ***
>
> > **[W1]** The experimental evaluation was only implemented in LLaMA-7B. How does it work on other mainstream models such as ChatGPT2?
>
> We conduct our experiment on GPT-2 pipeline, and the results on GSM-8K are given as follows:
>
> - Dropout rate is 0.2:
>
> |Methods|AdapEmu Accuracy|AdapFu Accuracy|
> |:-----:|:------:|:---------:|
> |Offsite Tuning|2.30%|2.06%|
> |FedOT|2.06%|2.30%|
> |FedBiOT (Adapter 2)|2.22%|2.14%|
> |FedBiOT (Adapter 4)|2.94%|2.54%|
>
> - Dropout rate is 0.5:
>
> |Methods|AdapEmu Accuracy|AdapFu Accuracy|
> |:-----:|:------:|:---------:|
> |Offsite Tuning|2.70%|1.59%|
> |FedOT|2.53%|1.51%|
> |FedBiOT (Adapter 2)|1.83%|2.06%|
> |FedBiOT (Adapter 4)|2.46%|2.54%|
>
> Although our proposed method outperforms the baselines in most settings, it is noticed that the performance of our proposed work is inferior to the baselines when the dropout rate is 0.5. In light that AdapFu falls behind AdapEmu in most cases, we hypothesize that GPT-2 is poor in math problem solving, and therefore, fine-tuning the model on the training data can perform better than aligning with the pre-trained full model.
>
> ***
>
> > **[W2]** In the experiment, federated learning only considered 8 clients. There is a lack of experiments that vary the number of clients and the number of training samples each client own.
>
> Please refer to Table 4 in Appendix B. We consider three different settings for the number of clients, i.e., three clients for math problem solving, nine clients for code generation, and eight clients for question answering. At the same time, the training samples vary among the clients when we fine-tune the model on code generation and question-answering tasks. Take a question-answering task (Dolly) as an example. The client with the OpenQA dataset holds 3611 samples, while the client with the creative writing dataset holds 711 samples.

---

### Official Review · Reviewer_uuim · 2023-11-03

**Soundness:** 3 good
**Presentation:** 3 good
**Contribution:** 2 fair
**Rating:** 3
**Confidence:** 5

**Summary:**

This paper proposes improvements to offsite tuning, a pre-existing large-scale federated learning model's partial tuning method. It is built upon the existing approach of offsite tuning, which involves splitting a transformer model into various sub-models by layer index, such as adapters and emulators. During FL training, clients receive a combination of adapters and emulators, with the emulator being frozen while the adapter is fine-tuned. This paper makes two key improvements. First, it selects the last few transformer layers as the adapter. Second, it introduces a public dataset on the server and reduces the KL divergence between the adapter-emulator and full model outputs through knowledge distillation.

**Strengths:**

1.	The paper is built upon a relatively recent work so that it may offer modern insights into the related research fields.

2.	Experimental results support the proposed improvements in the paper.

3.	The proposed improvements in the paper are general and should be easy to adopt.

**Weaknesses:**

1.	From a technical perspective, the two improvements proposed in the article may be incremental. One involves changing the index of the fine-tuning layers (based on observation), and the other relies on the traditional distillation method. Both methods are essentially at the level of tricks and are insufficient to serve as contributions to the paper.

2.	I have doubts about the "intellectual property protection" aspect of the paper. In this framework, although local clients can only obtain a portion of the model instead of the entire model, this sub-model can still be fine-tuned and used for inference, which implies that the majority of the model's functionality has been preserved. Essentially, malicious users can still steal this intellectual property. This framework does not seem to provide significant protection, so I do not consider the "intellectual property protection" mentioned in the title appropriate.

3.	The paper should provide a detailed algorithm to help readers follow.

**Questions:**

Weaknesses.

---

> ### Author Response · Authors · 2023-11-18
>
> We appreciate your time and efforts in reviewing our paper. Based on your comments, we would like to make the following clarification.
> ***
> > **[W1]** From a technical perspective, the two improvements proposed in the article may be incremental. One involves changing the index of the fine-tuning layers (based on observation), and the other relies on the traditional distillation method. Both methods are essentially at the level of tricks and are insufficient to serve as contributions to the paper.
>
> In our tasks, we want to overcome clients' computation constraints and protect both clients' data privacy and LLMs' intellectual property. Compared with the traditional distillation method, the data distribution used for alignment on the server differs from the downstream task; meanwhile, the clients fine-tune the trainable parameters on their end with heterogeneous task data. Thus, our task is significantly more challenging than the traditional knowledge distillation.
>
> To overcome such challenges, our key contribution to this paper is how we solve the challenges of LLM training in federated learning, i.e., formulating a bi-level optimization problem (Eq. 1 and 2). In detail, we expect AdapFu and AdapEmu to realize the outcomes that
> - **AdapEmu** is trainable for the clients and protects the original LLMs' privacy (i.e., model structure and parameters)
> - **AdapFu** boosts the performance on the downstream tasks by adopting some layers from **AdapEmu**
>
> Then, the problem is formulated accordingly, i.e., we disjoint an LLM into two parts (i.e., adapter and emulator) and align the knowledge between **AdapEmu** and **AdapFu** from round to round.
>
> To solve the formulated problem efficiently, we tried different layer selection strategies and noticed that the proposed one resulted in the best performance and the lowest computation costs.
>
> To the best of our knowledge, this is the **first** work that
> - trains a surrogate light model to boost the original full model using bi-level optimization.
> - makes LLM trainable on the clients without the data exposure to any third parties, in comparison with some recent works [1, 2, 3] focusing on LLM training in FL.
>
> **References:**
>
> [1] Towards Building the Federated GPT: Federated Instruction Tuning
>
> [2] Federated Tuning for Black Box Large Models
>
> [3] FedBPT: Efficient Federated Black-box Prompt Tuning for Large Language Models
> ***
> > **[W2]** I have doubts about the "intellectual property protection" aspect of the paper. In this framework, although local clients can only obtain a portion of the model instead of the entire model, this sub-model can still be fine-tuned and used for inference, which implies that the majority of the model's functionality has been preserved. Essentially, malicious users can still steal this intellectual property. This framework does not seem to provide significant protection, so I do not consider the "intellectual property protection" mentioned in the title appropriate.
>
> Let us rephrase your concern that the compressed model (AdapEmu) can reproduce the ability of the full model. It is an interesting issue that you consider from the functionality aspect, although our intellectual property protection focuses on the model architecture and part of the parameters. Below is the statement why our proposed method can protect the intellectual property in this case as well:
>
> It is evident that the compressed model (AdapEmu) performs much worse than the full model in terms of accuracy at the beginning and the end of the training. According to our experiments in Rosetta, the original LLaMA achieves a Pass@1 result of 10.5% for HumanEval, while the compressed LLaMA is of 0.61% when the dropout rate is 0.2. After 1000-round updates using Rosetta, although the compressed LLaMA boosts the performance to 5.61%, the result is still far behind the raw and updated full models (i.e., AdapFu, 12.68%). Apparently, our proposed method does not transfer the generalization property to the compressed model. From the business perspective, these two models with such performance gaps are *two different commodities*.
>
> Furthermore, even if the clients have the public datasets, it remains very difficult to reconstruct the full model. Given the existing understanding of LLM training, there is no guarantee that reconstructing the full model with the compressed one and the public dataset will succeed and cost less than retraining a full model from scratch. Thus, there is limited incentive for the clients with computation constraints in our setting to perform such attacks; therefore, the intellectual property of LLM is protected.
>
> Based on the above discussion, we assure that **the proposed FedBiOT provides sufficient protection for intellectual property**.
> ***
> > **[W3]** The paper should provide a detailed algorithm to help readers follow.
>
> Thank you for your suggestions. We make it available in Algorithm 1 of Appendix A, and a visualized demonstration is given in Figure 1.

---

> > ### Comment · Reviewer_uuim · 2023-11-23
> > **Thanks for the responses**
> >
> > Thank you for the rebuttal. I have read the authors' responses and therefore keep my rating.

---

### Author Response · Authors · 2023-11-21
**Look forward to your reponse as the discussion period ends in one day**

Dear area chair and reviewers,

Thank you for reviewing our paper and providing constructive feedback.

As today is the final day of discussion, we eagerly await your response and see if we address your comments. If you have more questions regarding our work, don't hesitate to point them out, and we are happy to discuss them with you.

Best,
Authors of paper 6725

---

### Meta-Review · Area_Chair_okhC · 2023-12-06

**Metareview:**

Due to privacy concerns, both data owners and owners of large language models (LLMs) are hesitant to share their data and models. This paper introduces FedBiOT, a method designed to address these concerns by ensuring data privacy while facilitating the fine-tuning of LLMs in federated learning tasks. The proposed approach formulates and solves a bi-level optimization problem to distill an emulator on a public dataset. This emulator supports local fine-tuning on private datasets without revealing the underlying LLM. Despite the author's rebuttal, one reviewer still perceives the reliance on traditional distillation methods as incremental, lacking significant contribution.

Concerning intellectual property protection, the reviewer contends that the author's rebuttal contradicts the content of the paper. In response, the author asserts that AdapEmu performs significantly worse than AdapFu. However, the reviewer points out that two terms in equation 2 seemingly encourage both AdapEmu and AdapFu to exhibit similar behavior, implying the author considers the output of AdapEmu beneficial to the full model. Furthermore, in Section 2.2, the author states, 'AdapEmu is adopted when users aim to achieve better generation results, e.g., solving a math problem.' This statement strongly suggests that AdapEmu serves a functional purpose.

Based on the recommendation from the reviewers, this paper needs further improvements before publication.

**Justification For Why Not Higher Score:**

N/A

**Justification For Why Not Lower Score:**

N/A

---

### Decision · Program_Chairs · 2024-01-16

Reject